# MAD: Move AI Decompiler to Improve Transparency and Auditability on Non-Open-Source Blockchain Smart Contract

## ABSTRACT

The vision of Web3 is to improve user control over data and assets, but one challenge that complicates this vision is the prevalence of non-transparent, scam-prone applications and vulnerable smart contracts that put web3 users at risk. While code audits are one solution to this problem, the lack of smart contracts source code on many blockchain platforms, such as Sui, hinders the ease of auditing. A promising approach to this issue is the use of a decompiler to reverse-engineer smart contract bytecode. However, existing decompilers for Sui produce code that is difficult to understand and cannot be directly recompiled. To address this, we developed the Move AI Decompiler (MAD), a Large Language Model (LLM)-powered web application that decompiles smart contract bytecodes on Sui into logically correct, human-readable, and re-compilable source code. MAD empowers developers to understand and audit contracts easily and independently.

Our evaluation shows that MAD produces logically correct code that successfully passes original unit tests and achieves a 66.7% recompilation success rate on real-world smart contracts. Additionally, in a user study involving 12 developers, MAD significantly reduced the auditing workload compared to using traditional decompilers. Participants found MAD's outputs comparable to the original source code, simplifying the process of smart contract logic comprehension and auditing. Despite some limitations, such as occasional hallucinations and compile errors, MAD still provides significant improvements over traditional decompilers.

MAD has practical implications for blockchain smart contract transparency, auditing, and education. It empowers users to review and audit non-open-source smart contracts, fostering trust and accountability. Additionally, MAD's approach could potentially extend to other smart contract languages, like Solidity, promoting transparency across various blockchains.

## CCS CONCEPTS

• **Human-centered computing** → **User interface toolkits**; • **Software and its engineering** → **Software reverse engineering**; • **Computing methodologies** → **Natural language generation**.

## KEYWORDS

Web3, Smart Contract, Auditing Tools, Large Language Models

## 1 INTRODUCTION

Web3, also referred to as the decentralized web, represents a significant shift in the evolution of the internet [20, 22]. Unlike earlier versions—Web1, which focused on static content, and Web2, which emphasized user-generated content and platforms—Web3 introduces decentralization, prioritizing user autonomy and ownership through blockchain technology [38]. A central element of Web3 is the use of smart contracts [29], which are self-executing, algorithmic-based agreements that automatically enforce terms. Smart contracts enable automated, trustless transactions, fostering the decentralized nature of Web3 ecosystems. This transition marks a major restructuring of internet architecture, shifting control from centralized entities to individuals and decentralized networks [37].

Nevertheless, the increased autonomy and control that Web3 grants to its users also demands that they take on greater accountability for their decisions within this decentralized ecosystem. Web3 has witnessed significant instances of fraud and vulnerabilities [5, 29], leading to substantial financial losses [7]. Notably, in the first half of 2024, over $1 billion in assets were lost in Web3-related cryptocurrencies, with phishing attacks (n = 150) and smart contract vulnerabilities (n = 105) being the most frequent causes [7]. In the absence of centralized regulators or platforms to verify contract security, one viable path is to empower users to audit smart contracts independently. By enabling users and communities to perform these audits, Web3 not only strengthens the transparency and accountability of decentralized applications but also reduces reliance on third-party authorities for security verification [1]. Independent auditing ensures that the decentralized ethos is upheld while mitigating risks inherent in open, trustless environments.

Unfortunately, on emerging blockchain platforms like Sui [28], smart contracts are published as bytecode, and their source code is not always open-source. This creates several potential risks, for example, the following are some real-world case studies:

(1) A user falls victim to **phishing**, losing 7 million USD worth of assets in a fake NFT whitelist register application.
(2) Developers deploy tokens while retaining minting rights within the smart contract, enabling the token to be minted indefinitely, leading to **unexpected inflation**.
(3) Decentralized exchanges contain **unexpected backdoors**, allowing users to withdraw assets deposited by other users.

If the smart contract source code were accessible, the aforementioned risks could be easily identified. However, these contracts are not open-source, and even if the source code is available on platforms like GitHub, users cannot be certain that it is the contract deployed on the blockchain. This underscores the critical need for **transparency** in smart contracts within the Web3 space. There is a need for a **effective, easy-to-use application** that allows users to independently **audit** the logic of non-open source smart contracts, improve **transparency**, and promote **algorithmic accountability** in the Web3 ecosystem. Our paper seeks to address this gap.

**Our study** developed and evaluated the Move AI Decompiler (MAD). MAD is a Large Language Model (LLM) powered web application that converts Sui Move smart contract bytecode into easily readable and re-compilable source code, enabling developers to understand and audit non-open-source smart contracts on Sui.

We summarized our main contribution as follows:

(1) Developed and evaluated MAD, an LLM-powered decompiler that generates logically accurate, human-readable, and re-compilable Move code from bytecode. Achieved a 66.7% recompilation success rate, promoting transparency and ease of auditing for non-open-source smart contracts.

(2) Conducted a user study with 12 developers, demonstrating that MAD's output has similar workloads to working with source code and has a significantly lower reading workload compared to existing machine-based decompiler.

(3) The user study provided insights that highlight MAD's practical implications in promoting transparency and trust in Web3, addressing security concerns, and facilitating contract auditing.

## 2 BACKGROUND AND RELATED WORK

### 2.1 The Move Programming Language

In the current blockchain ecosystem, Solidity [36] remains the predominant language for smart contract development [22]. However, it is also prone to vulnerabilities such as reentrancy attacks and integer overflows [39], which have been exploited by malicious actors, leading to significant financial losses [5, 7, 29]. In response to these security concerns, several new blockchain platforms have chosen to develop new domain-specific languages aimed at addressing the shortcomings of Solidity. For instance, emerging blockchains like Sui [28], Aptos [10], Diem [27], and Movement [14] have adopted the Move language [3] for smart contract development.

Move language [3], originally developed for Meta's Diem (formerly Libra) project [27], was designed to address many vulnerabilities found in Solidity by incorporating several key features. Specifically, Move offers the following advantages to enhance security and prevent common vulnerabilities:

**Resource Management and Safety**: Unlike Solidity, which lacks built-in resource management and requires developers to prevent issues like reentrancy attacks or resource leakage manually [39], Move treats assets as first-class resources. It enforces strict rules through its type system to prevent accidental creation, duplication, or destruction of assets. For instance, when an object is passed by value to a function in Move, it becomes frozen and cannot be reused unless explicitly handled. This mechanism eliminates vulnerabilities such as reentrancy attacks by ensuring that resources are managed safely and predictably.

**Static Verification and Error Detection**: Solidity often relies on external tools to catch errors and security vulnerabilities, which means issues may only be discovered at runtime [39]. In contrast, Move employs a strong static type system and allows for formal verification during compilation. This ensures that potential errors and vulnerabilities are detected early in the development process, reducing the chances of runtime security risks.

Despite the significant advantages offered by the Move language over Solidity, Move remains relatively immature and lacks comprehensive learning resources. The novel paradigms introduced by Move present a learning curve for developers unfamiliar with its structure and conventions, potentially leading to logical errors and, consequently, security vulnerabilities such as access control issues. Therefore, code reviews and safety checks are essential for thoroughly evaluating Move smart contracts to ensure they are free from logical flaws and other security risks.

Furthermore, Move contracts can still conceal malicious components, including phishing mechanisms, hidden backdoors, and other logic designed to disadvantage general users.

Additionally, in the Move ecosystem, most contracts are not open-source. Taking Sui Move as an example, over three-quarters of the top Sui projects on DefiLLama [8] have not provided their source code. Similarly, almost none of the Coin and NFT projects on Sui [31, 32] have provided source code. This lack of **transparency** makes it extremely difficult for the public to **audit** deployed Move smart contracts. Consequently, users are unable to verify the security and reliability of these contracts, hindering their ability to use them with confidence.

### 2.2 Decompiler on Move Language

To address the limited open-source availability of Move contracts in the Sui ecosystem, the Revela Decompiler [30] and Move Disassembler [17] were developed. However, despite their potential, both tools have two limitations that discourage users.

Firstly, like other decompilers, it can only generate variable names such as v0, v1, and v2, as illustrated in Figure 1. The missing variable names make it difficult for users to perform thorough code reviews on decompiled contracts as it is hard to interpret.

```
Revela Decompiler                          Source Code
public fun create_project<T0>(              public fun create_project<T>(
  arg0: &mut 0x2::tx_context::TxContext       ctx: &mut TxContext,
) {                                         ) {
  let v0 = Project<T0>{                        let project = Project<T> {
    // Error for invalid object creation         id: object::new(ctx),
    // without a newly created UID.             };
    // need to use sui::object::new instead.
    id: 0x2::object::new(arg0)                 let project_id = object::id(&project);
  };
  // `v0` is moved to the share object        transfer::share_object(project);
  // and cannot be access anymore
  0x2::transfer::share_object(v0);            let event = ProjectCreatedEvent {
  // Compile Error: cannot use `&v0` here       project_id
  // because `v0` is moved                    };
  let v1 = ProjectCreatedEvent {
    project_id: 0x2::object::id(&v0)           event::emit(event);
  };                                         }
  0x2::event::emit(v1);
}
```

**Figure 1: Example illustrated how Revela's output would yield errors when re-compile with Sui Move compiler.**

Second, the output from Revela and Move Disassembler cannot be directly recompiled due to the complex rules of the Move compiler. For example, Revela struggles with managing ownership constraints, preventing the access of an object while it is being mutated within the same line, and handling frozen objects, as illustrated in the code comments in Figure 1. These issues stem from the Move language's strict resource and ownership rules, which are challenging to decompile accurately. The inability to recompile the decompiled code hinders further verification of security properties and the detection of potential vulnerabilities, such as running unit tests to assess whether the smart contract, when interacting with other deployed contracts, could introduce potential vulnerabilities.

To address this issue, an effective decompiler that allows users to inspect the source code of smart contracts easily and produce re-compilable code is needed. This raises our first research question:

**RQ1:** Can we decompile Move bytecodes into code that is logically equivalent to the original, and can it be successfully recompiled?

By investigating RQ1, we aim to determine whether it is feasible to generate decompiled code that maintains the original logic and functionality, thereby enhancing transparency and trust in the Sui Move ecosystem.

## 2.3 AI Augmented Decompiler

Decompilers can never fully reconstruct the original developer-written code [25]. Vital elements such as comments, variable names, and types, which significantly contribute to program comprehension [11, 15], are typically absent from decompiler output.

Recent research has explored the use of artificial intelligence (AI) and machine learning, particularly Large Language Models (LLMs), to augment decompiler output [25, 26, 35] to make it easier to read and ready to compile. LLMs like OpenAI's GPT-4 have demonstrated remarkable capabilities in code understanding and generation, even with minimal task-specific training data [4]. For instance, [26] fine-tuned GPT-4o to significantly outperform traditional decompilation tools on C language, achieving a 74.3% re-executability rate on the HumanEval benchmark [6]. These results indicate that LLMs can enhance the effectiveness of decompilation by generating more comprehensible and executable code.

However, AI-augmented decompilers have primarily focused on mainstream programming languages like C [6, 25, 26, 35], which have extensive data for training and evaluation. Their success depends on large labeled datasets, making them less suitable for newer, domain-specific smart contract languages like Move that lack training data. The scarcity of Move code limits the effectiveness of traditional fine-tuning methods that require substantial training data. Recognizing these limitations, it becomes essential to explore alternative approaches that do not heavily rely on large, labeled datasets.

One promising avenue is leveraging the prompt engineering [23, 24] and few-shot learning [4] capabilities of LLMs. Due to their extensive pre-training on vast amounts of text data—including code from various programming languages—LLMs can generalize to new tasks with limited examples in the prompt [4, 34]. This adaptability makes them particularly well-suited for decompiling emerging domain-specific languages like Move. By employing LLMs in this context, we aim to overcome the data scarcity issue and enhance both the readability and executability of decompiled Move code.

It is particularly noteworthy that since the Sui mainnet was launched in May 2023, some LLMs don't have any training data about Sui Move, for example, *GPT-4-1106-preview* only have data until April 2023. This raises a very interesting research question:

**RQ2**: How stable are LLMs in decompiling Sui Move, and how does their pre-trained knowledge of Move affect their ability to generate re-compilable Move code?

By investigating this question, we aim to examine how LLMs' prior knowledge influences their ability to decompile domain-specific languages like Move. Additionally, our research seeks to generalize these findings to other domain-specific smart contract languages, ultimately fostering greater **transparency** and the ease of **auditability** of smart contracts across various Web3 ecosystems.

## 2.4 Expectancy Theory and Auditing

Expectancy Theory [21, 33] is a motivational theory that explains the decision-making process individuals use to pursue certain actions based on the expectation of desired outcomes. The theory suggests that individuals are motivated to act around three key components:

- **Expectancy**: The belief that one's effort will lead to the desired level of performance.
- **Instrumentality**: The belief that achieving the performance will lead to specific outcomes or rewards.
- **Valence**: The value or importance the individual places on the expected reward.

In the context of smart contract auditing, Expectancy refers to the users' belief that they will be able to understand the code effectively. When faced with non-open-source contracts or the limitations of decompilers like Revela, this belief is weakened, reducing their motivation to engage. Instrumentality reflects how users perceive their efforts will result in desirable outcomes, such as detecting vulnerabilities or increasing contract security. Valence, on the other hand, refers to the value users place on these outcomes—how important it is to them to ensure transparency, security, and fairness in smart contracts.

MAD directly addresses these concerns by generating human-readable and recompilable code, increasing users' expectancy of successfully understanding the smart contract logic. As a result, users are more likely to perceive their auditing efforts as instrumental in achieving meaningful, secure results. The higher the Valence users assign to these outcomes—whether it's protecting their assets or contributing to a more secure ecosystem—the more motivated they are to use tools like MAD to audit contracts thoroughly.

Thus, our third research question (RQ3) seeks to explore how Web3 users perceive the outputs of the MAD decompiler:

**RQ3:** How do Web3 users perceive the output of MAD Decompiler, and how do they intend to use it?

By addressing RQ3, we aim to understand whether the MAD enhances users' ability to comprehend smart contracts and promotes greater trust and transparency in the Web3 ecosystem.

## 3 DEVELOPMENT OF MOVE AI DECOMPILER

MAD leverages the outputs of the Revela Decompiler and the Move Disassembler with prompt engineering techniques to feed into large language models (LLMs), aiming to generate human-readable and re-compilable code.

Intuitively, one might consider feeding the entire output from Revela directly into the LLMs and instructing them to generate the complete code. However, this approach is not feasible in practice. We observed that LLMs struggle to handle long code inputs and often omit parts of the code by summarizing them as comments (see Figure 6 in appendix) or even hallucinations [2] like omitting or inventing functions. Therefore, it is necessary to process the code in smaller chunks to achieve the desired output.

In our approach, we split the chunk on a per-function basis; then, the input is fed into the LLM using a carefully engineered prompt. We construct our prompt into the following components:

(1) **Domain-specific knowledge**: We input specific knowledge of Sui Move into the LLM, including language features, syntax, variable mutation, and object ownership. We also provide instructions on errors commonly encountered in Revela's output, guiding the LLM to fix these issues during conversion.

(2) **Should and should not instructions**: These instructions emphasize the LLM's task while avoiding common mistakes,

such as ensuring output is well-formatted, using clear variable names, including all necessary type annotations, and should not having hallucinations.

(3) **Few-shot examples**: By providing function code for the input with Revela decompiler, along with the output of the original source code, the LLM is trained to understand the expected input and output format and the syntax of Sui Move. We deliberately selected 17 diverse examples to ensure coverage of the most common scenarios.

Our prompt, after times of iteration, contained 36,120 characters after stringified. With this well-crafted prompt, we ensured that the model could learn various aspects of the Move language and provided enough examples for it to output well-formed Move code. Our full prompt is available at [1]. At the time of development, we only had access to GPT-4 with a knowledge cut-off date of April 2024, which did not train on any data about the Move language with Sui. Although we also attempted fine-tuning with *GPT-3.5-turbo*, which is the only model we can fine-tune with at the time, the results were suboptimal, as the model was more prone to hallucinations.

We developed both the frontend and backend of MAD using Next.js and Vercel. By utilizing Vercel's Edge Functions, MAD efficiently processes function chunk decompilation requests in parallel, enhancing overall speed. Users with an API key or wallet on Sui can easily decompile smart contracts by providing the contract ID through the MAD web application. The anonymous platform link is [2], and the anonymous source code can be found at [3].

## 4 SYSTEM EVALUATION METHODS

To validate the effectiveness of MAD, we created a comprehensive evaluation framework that assesses its performance on both examples with unit tests and real-world smart contracts. This approach tests MAD's ability to handle varying levels of complexity, ensuring reliability in practical applications. The full list of packages we used to evaluate and the evaluation script is available at our GitHub.

When evaluating, the OpenAI API's temperature parameter is set to 0, and the seed is set to "123" to ensure the best reproducibility.

### 4.1 Evaluation on Examples with Unit Test

We first verified that our pipeline could decompile Move bytecodes into logically equivalent source code that can be successfully recompiled, executed, and pass unit tests. We utilized 10 example packages with unit tests from Sui Move's official repositories version 1.22.0 [4]. These examples provide a known baseline for assessing the functionality and correctness of the decompiled code. We deployed these smart contracts, used MAD to decompile them, and then checked whether the decompiled code could be recompiled and pass the unit tests in the original source code.

### 4.2 Evaluation on Real-world Contracts

To further evaluate MAD in real-world settings, we decompiled the top 30 real-world smart contracts without third-party dependencies,

selected from Sui Explorer [5]. The exclusion of dependencies was necessary to focus on the contracts themselves, as dependencies add extra complexity to the evaluation pipeline. The contracts chosen represent a diverse set of categories, including gaming, decentralized exchanges and marketplace, Non-Fungible Tokens (NFTs), .etc.

## 5 SYSTEM EVALUATION RESULTS

### 5.1 Result on Examples with Unit Test (RQ1)

The results from our evaluation of smart contracts with unit tests show that **60%** of the 10 example contracts decompiled by the MAD Decompiler using *GPT-4o-2024-08-06* were able to be recompiled and pass unit-test successfully without any modification. The remaining 40% of contracts were logically correct but encountered some Move language rule check errors inherited from Revela's output, such as using variables after they are frozen. After manually addressing these issues, the unit tests for those contracts also passed [6].

These results indicate that MAD is capable of generating code that is logically equivalent to the original source code. However, even though MAD does fix some issues for Move language rules from Revela's output, minor modifications may sometimes be necessary to ensure successful recompilation and execution.

### 5.2 Results on Real-world Contracts (RQ2)

In the evaluation of real-world contracts, we observed that MAD, utilizing *GPT-4o-2024-08-06* (with training data until October 2023), successfully generated outputs that can be recompiled in **66.7%** of cases. In comparison, *GPT-4-1106-preview* (with training data until April 2023) and *GPT-4-0125-preview* (with training data until December 2023) both achieved a recompilation success rate of **53.3%**. This indicates that the pre-trained knowledge of the model doesn't influence its decompilation performance, but larger models might perform better on this task. Although all versions generated logically sound code, errors related to Move language rule checks were observed. Furthermore, we identified some hallucinations. For example, the model incorrectly substituted a custom vector range-checking function with a built-in vector function instead of using the internal function defined in the smart contract. The full output and error of these models can be found in our GitHub [7].

These findings suggest that while MAD enables LLMs to generate logically correct Move code, some syntax errors and hallucinations are still unavoidable, especially in complex real-world contracts. However, the results are good enough to enhance **transparency**, allowing users to examine the logic of the contracts themselves. Therefore, we conducted a user study to evaluate this aspect.

## 6 USER STUDY DESIGN

The ability of MAD to generate decompiled code that is both logically correct and human-readable positions it as a valuable tool for users looking to audit closed-source smart contracts. To investigate its potential further, we invited 12 web3 users with different levels of experience in Sui Move language development to participate in our user study. Our study was approved by the IRB.

---

[1]https://reurl.cc/MjmyjW
[2]https://move-ai-decompiler.vercel.app/decompile?api_key=WWW25
[3]https://github.com/anonymousStars/MAD_WWW
[4]https://github.com/MystenLabs/sui/tree/c490f3a19447d17c96cd664729ad39fef32b7230/examples/move

[5]https://suivision.xyz/packages, data extracted on September 20
[6]The decompiled contract for unit test can be found at https://github.com/anonymousStars/MAD_WWW/tree/main/unit_test_gpt-4o_decompiled
[7]https://github.com/anonymousStars/MAD_WWW/blob/main/evaluation.ipynb

## 6.1 90 minutes user study for code reading (P)

7 participants engaged in a 90-minute session where they worked on a code comprehension task and a 15-minute vulnerabilities detection session for a smart contract related to a staking and voting campaign [8]. The smart contract was created by an experienced Move developer and included 6 critical vulnerabilities such as access control, function visibility, and backdoor function. These vulnerabilities were inspired by real-world examples of security flaws that have been observed in deployed Sui Move smart contracts.

Participants were randomly assigned to use either Revela (n = 3) or the MAD (n = 4) for the code comprehension task within the same interface; they answered questions such as "Explain the logic of function X," during which their completion times were measured. Following this, they engaged in a 15-minute vulnerability detection task, where the number of vulnerabilities they identified was recorded. After completing the tasks, participants reviewed the correct answers in the vulnerability detection task, code in alternative conditions, and original human-written source code. Then, they participated in a semi-structured interview, during which they filled out the NASA-TLX survey and explored the MAD web applications. The interface screenshot of the experiment is at Figure 8.

## 6.2 30 minutes Quick Interview (Q)

Additionally, 5 highly experienced Sui Move developers took part in a shorter, 30-minute study. They used the MAD to decompile their own smart contracts, reviewed the decompiled contract code, and compared the output from Revela, MAD, and their own source code. After that, they completed the NASA-TLX survey along with a semi-structured interview.

## 6.3 Measures of readability

We used a 7-point Linkert scale for the NASA Task Load Index (NASA-TLX) [12, 13] to enable participants to evaluate their overall workload while performing code reading and bug-catching tasks under various conditions. NASA-TLX is a widely adopted tool for assessing perceived workload across six dimensions: mental demand, physical demand, temporal demand, effort, performance, and frustration. This provides a comprehensive measure of both cognitive and physical strain experienced during a task, offering a deeper insight into task difficulty and workload [12, 13].

Participants in 90 minutes study group (P) were randomly assigned either decompiled code produced by Revela or by the MAD, followed by code comprehension and vulnerability detection tasks. We compared their task finish time between different conditions. After completing the tasks, they were asked whether they thought the code (MAD or Revela's output) they engaged with was human-written source code or not. Then, they complete the NASA-TLX questionnaire for their assigned condition. They were then prompted to consider whether their NASA-TLX scores might differ under alternative conditions or when reading the human-written code.

As for the Quick Interview Group (Q), participants were asked to evaluate the workload of using Revela, MAD, and source code when auditing Move contract code with NASA-TLX with their own deployed smart contract.

[8]The code is available at https://reurl.cc/93AR3a

## 6.4 Semi-structured interview

During the completion of NASA-TLX, we engaged participants in discussions to understand the rationale behind their choices. For instance, we asked questions like, "You rated Revela higher than MAD on Mental Demand—could you explain why?"

After completing the NASA-TLX, we asked participants open-ended questions to gain further insights into the potential, limitations, and concerns of MAD. These questions included:

(1) What are your overall thoughts on MAD?
(2) How and why you might use MAD in the future?
(3) Do you have any concerns about MAD?
(4) How would you feel if the smart contract you deployed was being decompiled by others by MAD?

We applied a thematic analysis [9] approach to analyze the interview transcripts, focusing on extracting insights from participant responses. As suggested by [16], thematic analysis is well-suited for understanding participant perspectives in exploratory studies. Given the qualitative nature of this analysis, agreement scores were not deemed necessary, as the primary goal was to capture a wide range of user insights without the need for multiple coders.

## 6.5 Other Informal Conversations

The authors also engaged in informal discussions with several Web3 project leaders and managers at in-person events or by message to gather their opinions on MAD. In particular, we asked about concerns raised by participants in semi-structured interview sessions.

## 7 USER STUDY RESULTS (RQ3)

### 7.1 Perception of Decompiled Code

When asked, all 3 participants in the Revela condition consider the code is not human-written source code, whereas 75% (3 out of 4) of participants in the MAD condition believed the code they read was source code [9]. This suggests that MAD's decompiled code feels like actual source code to the participants.

### 7.2 Comparative Analysis of NASA-TLX Scores

The result of NASA-TLX is shown at Figure 2. Statistical analyses showed significant higher workload in all dimensions between Revela and the other two conditions (all $p < .01$), with no significant differences between MAD and original source code. This suggests that MAD provides a reading and auditing workload comparable to the source code. The Cronbach's Alpha for the NASA-TLX scores across all dimensions was 0.843, indicating good internal consistency. The detail of the statistical analysis results is at Appendix A.

### 7.3 User Interview Result on NASA-TLX

*7.3.1 Mental, Physical, Temporal Demand and Effort.* All participants considered that MAD's variable names were easier to understand compared to Revela's, requiring less mental effort. Additionally, because the variable names were more intuitive, there was less need for scrolling around, reducing the physical demand. The ease of comprehension also lowered temporal pressure, making the

[9]We believe the one participant in the MAD group who thought it was not human-written source code misunderstood the question, as he found some bugs in the code and assumed it must have been AI-generated because of these bugs.

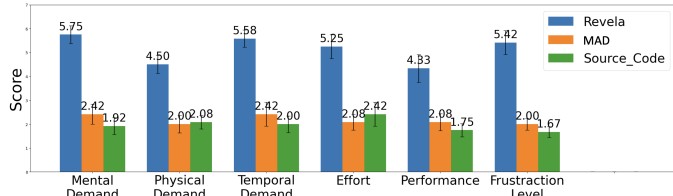

**Figure 2: NASA-TLX results for users across Revela, MAD, and Source Code condition. Lower is better.**

overall effort lighter. Two participants from the 90-minute group (P) even found MAD's format and variable choices easier to understand than the original source code itself. For example, as illustrated in Figure 3, one participant noted that MAD's use of the variable name "exist" was clearer than the source code's "is_in_leaderboard".

```
let (v0, v1) = 0x1::vector::index_of<0x2::object::ID>(&arg0.top_projects, &arg1);
if (v0) {
    0x1::vector::remove<0x2::object::ID>(&mut arg0.top_projects, v1);
    0x1::vector::remove<u64>(&mut arg0.top_balances, v1);
};                                                      Revela

let (exists, index) = vector::index_of(&leaderboard.top_projects, &project_id);
if (exists) {
    vector::remove(&mut leaderboard.top_projects, index);
    vector::remove(&mut leaderboard.top_balances, index);
};                                                      MAD

let (is_in_leaderboard, index) = vector::index_of(&leaderboard.top_projects, &project_id);
if (is_in_leaderboard) {
    vector::remove(&mut leaderboard.top_projects, index);
    vector::remove(&mut leaderboard.top_balances, index);   Source Code
};
```

**Figure 3: Example output illustrated the difference between Revela, MAD, and Source Code's output.**

However, in the quick interview group (Q), participants mentioned that after decompiling their own contracts, MAD's output was not as polished as the original source code. This gap was due to the absence of developer-written comments and advanced Move language features such as Macros [18], which define reusable code snippets, and Method Syntax [19], which allows functions to be invoked directly from variables. These elements are optimized during compilation but not reflected in the decompiled output. Despite this, participants felt that these differences only lead to a more redundant structure but do not hinder readability or auditing.

These results indicate that MAD can effectively generate smart contract code that closely resembles the original source code, making it readable and suitable for auditing the underlying logic.

*7.3.2 Frustration Level.* Compared to the source code or MAD's code, participants generally felt more frustration when reading code from Revela. The lack of variable names was especially discouraging, making it harder for users to audit smart contract codes. One participant even admitted during the interview that they almost wanted to give up the experiment while working with the Revela condition. In contrast, participants reading under the MAD condition found it much more intuitive and easy to read, just like reading the original source code.

*7.3.3 Performance.* For the performance dimension in NASA-TLX, all participants perceived their code comprehension and auditing performance to be better in the MAD or Source Code conditions

than in the Revela condition. However, for actual performance, we found the number of identified vulnerabilities in the MAD (M = 2.00, SD = 0.82) and Revela (M = 1.33, SD = 1.53) conditions did not differ in a statistically significant manner (t(5) = 0.76, p = .48). Additionally, within a 15-minute time limit, there were 6 vulnerabilities in total, and no participant was able to identify more than half of them.

Additionally, the time spent on function comprehension task showed a similar pattern, with MAD (M = 291.75, SD = 157.86) requiring less time than Revela (M = 450.33, SD = 182.40). However, this difference was not statistically significant (t(5) = 1.24, p = .27), and both conditions took a considerable amount of time.

Notably, after the task, when users viewed the task answers, all participants completed checking and validating all six vulnerabilities within 3 minutes because the answers were directly commented on where the vulnerabilities appeared. The results indicate that while MAD's code is indeed easier to read, users may still need more time or additional assistance to effectively identify vulnerabilities in smart contract code. We will discuss more about this in the future work discussion section 8.3.2.

## 7.4 Semi-Structured Interview Results

We asked participants open-ended questions to gain further insights into the potential, limitations, and concerns of MAD.

*7.4.1 Usefulness of MAD.* All participants found MAD to be a highly useful tool. Many praised its ability to generate decompiled code that closely resembled the original source code, making it easier to read and understand. For example, **P1** mentioned that MAD's output was "quite readable and comparable to source code," which improved their ability to analyze already deployed smart contracts. Similarly, **Q1** appreciated how MAD made the decompiled code "much more readable," which helped them better understand the code structure.

The ability to analyze non-open-source contracts was a significant advantage noted by participants. **P4** expressed that MAD would be useful for examining non-open-source projects, meme coins, and decentralized applications to ensure safety before investing. **P7** highlighted that they could use MAD for auditing purposes, such as decompiling contracts to identify bugs in projects they are considering for investment. **Q2** pointed out that MAD would be valuable for analyzing specific cases on the network, especially when they have no direct access to the code or its authors.

Moreover, MAD was also appreciated for development purposes. **P6** saw value in using MAD to check other people's code and incorporate relevant sections into their own projects. **Q5** highlighted that MAD made it easier to understand and debug non-open-source contracts by revealing the internal logic more clearly.

Several participants emphasized MAD's usefulness in the educational context. **P2** mentioned that MAD could be a tool for learning purposes, allowing them to learn from well-known production smart contracts and improve their own development skills. **P3** saw MAD as helpful for reverse engineering and learning from other developers' code, aiding in debugging and enhancing their own smart contracts. **Q4** echoed this sentiment, noting that MAD would be highly beneficial for understanding other protocols' code, especially for developers new to Move programming.

*7.4.2 Concerns About MAD.* While participants generally appreciate MAD, some concerns were raised about its potential misuse. **P5** was worried that MAD could make any project effectively open-source, which could be seen as a threat by developers who want to keep their code private. They also noted that attackers might use MAD to decompile smart contracts and find vulnerabilities. **P6** similarly expressed concerns about competitors using MAD to clone proprietary contracts, highlighting the risk of intellectual property theft. **P5** also noted that the ease of access to code could pose risks, particularly if attackers use the decompiled code to exploit vulnerabilities.

Other participants were concerned about the accuracy of MAD's decompilation. **P1** mentioned that while they had not encountered major issues yet, they were still cautious about the possibility of "hallucinations, such as incorrect or made-up code", which could erode their trust in the tool.

Despite these concerns, many participants valued the transparency MAD offers. **P3** stated that transparency is essential in blockchain, as it allows everyone to verify that no malicious actions are being hidden in smart contracts.

*7.4.3 Concerns About Their Own Code Being Decompiled.* Participants had mixed feelings about someone else decompiling their own code by MAD.

Several participants were comfortable with it, as they valued transparency and openness principles of Blockchain. **P3** expressed no concerns, explaining that if their code was free of vulnerabilities, they had nothing to worry about. **P4** said they were fine with their code being decompiled, as it aligned with the open nature of blockchain technology. **Q1** shared a similar view, stating that they preferred open-source practices and saw transparency as key to blockchain's philosophy.

Moreover, Some participants, such as **Q3**, were indifferent to having their code decompiled because their smart contracts were already open-source. **Q4** also welcomed the idea, mentioning that decompiling their code could make it easier for others to understand, audit, and even learn from it.

However, a few participants expressed discomfort with the idea of their code being decompiled. **P5** admitted they would feel uneasy if their non-open-source code were decompiled, fearing that it could expose their project to risks if vulnerabilities were discovered. **P6** also raised concerns about competitors potentially cloning their proprietary contracts using MAD, which could lead to intellectual property theft.

**Q2** and **Q5** noted that all on-chain code is public, meaning determined individuals can reverse engineer it with enough effort. They explained that MAD simply streamlines this common process, making code more accessible for analysis, which aligns with blockchain's open nature. **Q5** added that developers should expect contracts to be open sourced and analyzed once deployed, viewing decompilation as part of the norm rather than a major issue.

In summary, while some participants had concerns about MAD's potential misuse, most were supportive of the transparency it brings to the blockchain space. Many embraced the idea of their code being decompiled as part of maintaining a transparent Web3 ecosystem.

## 7.5 Insights from Discussion with Web3 Projects

We engaged in informal discussions with several Web3 project leaders and managers about the concerns raised from our interview result at subsubsection 7.4.2 and 7.4.3.

It's worth mentioning that during these conversations, one project leader emphasized how their team had used MAD to accelerate the integration of external smart contracts, particularly when verifying the correctness of complex logic, saving weeks of research time. This demonstrates MAD's potential to simplify integration processes in Web3 projects, offering algorithmic transparency and promoting both technical efficiency and operational benefits.

*7.5.1 Concerns about Exposing Vulnerabilities.* Regarding the concern that MAD might expose vulnerabilities in their smart contracts, project leaders and managers generally expressed confidence. Many highlighted that their contracts had undergone professional security audits, which serve as a strong endorsement of the contract's security. They pointed out that if a hacker possessed the capability to find vulnerabilities that these audit teams missed, such a hacker would likely be able to reverse-engineer the contract logic regardless of the availability of tools like MAD.

When asked directly about the concerns of people decompiling their contracts, one owner viewed the contracts as effectively open-source once deployed on the blockchain. Their reasoning was that, by being accessible on the blockchain, anyone with enough knowledge could theoretically extract the contract's logic, regardless of MAD's existence. Thus, MAD merely speeds up a process that would otherwise take considerable time and expertise.

*7.5.2 Concerns about Competitors Cloning Contracts.* As for concerns related to competitors potentially cloning their smart contracts using MAD, project owners expressed that merely decompiling the smart contract does not provide a significant advantage. They emphasized that a successful Web3 project relies on much more than its contract alone. Other critical factors include frontend development, backend infrastructure, marketing, and the liquidity value provided to the project. In sum, they noted that contracts are just one part of a much larger, complex system that cannot be easily replicated without significant effort and investment.

Regarding intellectual property issues, one project owner mentioned that their close-sourced smart contract was protected under a Business Source License (BUSL), meaning that even others decompiled it, it still have legal protection. Nevertheless, upon further investigation, we found no explicit BUSL declaration for their project. Since the legal issues of the use of decompiled code is outside the scope of this paper, we will not discuss this matter further.

## 8 DISCUSSION

Our study highlights the promising potential of MAD as an AI-powered decompiler for improving transparency and auditability in the Web3 ecosystem. By addressing the non-transparent issues posed by non-open-source smart contracts, MAD enables web3 developers to independently audit blockchain contracts. This significantly enhances users' ability to detect vulnerabilities and ensures the integrity of Web3 applications.

## 8.1 Practical Implications

The practical implications of using LLMs to decompile smart contracts extend beyond code readability. By offering a decompilation solution that translates bytecode into logically equivalent and human-readable code, MAD makes blockchain auditing more accessible. This is crucial in the context of blockchain financial and security applications, where trust and transparency are paramount. Our findings indicate that participants have less workload in understanding the code decompiled by MAD compared to existing decompilers like Revela. This suggests that tools like MAD can empower Web3 developers, along with other users, to identify potential risks more easily before engaging with smart contracts.

Moreover, MAD's application in real-world use cases, such as auditing closed-source contracts, introduces a powerful tool for preventing exploitation and fraud. By empowering the developer community to inspect the logic in smart contracts, MAD promotes a more transparent and auditable ecosystem where malicious code is more easily detected. This can help Web3 users avoid vulnerable smart contracts, such as phishing, backdoors, and unfair logic.

In addition to improving transparency, MAD has the potential to be a developer education tool. As noted by several participants, MAD could be used to reverse-engineer non-open-source contracts for learning and development purposes. This educational value offers an opportunity for developers, particularly those new to the Move language, to learn from real-world production codes.

Another key implication is MAD's potential generalizability to other domain-specific smart contract languages, regardless of whether the LLM has been trained with knowledge related to those languages or whether training data is available. With the right prompt engineering technique, the same pipeline used in MAD can be adapted to decompile other languages such as Solidity and Rust. This approach would extend AI-powered decompilation, enhancing transparency and auditability across various Web3 ecosystems.

## 8.2 Limitations

Despite its advantages, MAD still has limitations. Our study identified some minor LLM hallucinations in MAD's decompiled code, such as incorrect function substitutions. While these hallucinations did not significantly hinder comprehension, they may limit the tool's effectiveness in complex contracts.

Furthermore, although MAD addresses some errors from the Revela decompiler input related to syntax and resource constraints in Move, manual intervention is still needed to ensure that decompiled code complies with Move's strict compiler rules and can be successfully recompiled.

## 8.3 Future Directions

### 8.3.1 Improving Support for Advanced Move Language Features.
Our future work involves enhancing the decompiler's ability to handle advanced Move language features, such as macros and method syntax, to increase its accuracy and utility in real-world scenarios. Fine-tuning new LLMs such as *GPT-4o* specifically for Move and improving the integration of Move-specific knowledge into the decompilation process could further reduce hallucinations and improve the decompiled output's accuracy.

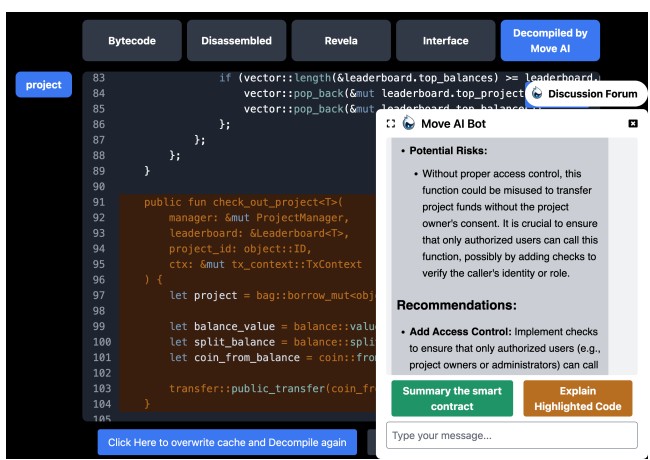

**Figure 4: Screenshot of current MAD interface, where users can chat with a AI chatbot about potential vulnerabilities**

### 8.3.2 Enhancing AI and Community Collaboration in Vulnerability Detection.
Our study showed that within a 15-minute time limit, participants with Move development experience were only able to identify fewer than half of the six critical vulnerabilities embedded in the smart contracts. Nevertheless, after being shown the task answers, all participants were able to check, validate, and understand all six vulnerabilities within three minutes. These findings suggest that while MAD's decompiled code is indeed easier to read, developers may still need additional assistance to effectively identify vulnerabilities in the decompiled code.

This raises an important opportunity: enabling AI and communities to identify potential vulnerabilities and allowing users to make the final judgment on these findings. To explore this further, we have already integrated an AI-powered chatbot and a community forum into the MAD web application, as shown in Figure 4 and Figure 7 in Appendix B. The AI-powered chatbot feature enables developers or even non-technical users to discuss the vulnerabilities of a given non-open-source smart contract with AI. Meanwhile, the community forum allows users to discuss specific contracts, collaboratively point out vulnerabilities, and share audit reports.

In our preliminary tests with this functionality, we found that AI was able to identify most of the vulnerabilities with carefully crafted prompts, but it also generated several false alarms. Therefore, our future work will focus on improving the collaboration between humans and AI, reducing false positives, and encouraging users to share audit results at the community forum. This exploration aims to enhance the role of AI as an auditing assistant while ensuring that human oversight remains central to the auditing process. By fostering collaboration between humans, AI, and the community, we hope to further streamline the process of identifying and addressing vulnerabilities in Web3 smart contracts.

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

## A DATA ANALYSIS DETAIL RESULTS

The analysis aimed to compare the perceived workload of participants using three different code versions: Revela, MAD, and the Source Code. The NASA Task Load Index (NASA-TLX) scores were collected across six dimensions: Mental Demand, Physical Demand, Temporal Demand, Effort, Performance, and Frustration Level.

### A.1 Comparison of NASA-TLX Scores

One-way ANOVA tests were conducted for each dimension to determine if there were significant differences among the three conditions. The results indicated significant differences across all dimensions (all $p < .001$). Post-hoc Tukey's HSD tests revealed that both MAD and Source Code conditions had significantly lower workload scores compared to the Revela condition, while there were no significant differences between the MAD and Source Code conditions.

*A.1.1 Mental Demand.* Participants reported lower mental demand when using MAD (M = 2.41, SD = 1.44) and Source Code (M = 1.92, SD = 1.24) compared to Revela (M = 5.75, SD = 1.29). The ANOVA showed a significant effect of the code version on mental demand, ($F_{(2, 69)} = 29.61$, $p < .001$). Post-hoc tests confirmed that both MAD and Source Code scores were significantly lower than Revela (both $p < .001$), with no significant difference between MAD and Source Code ($p = .63$).

*A.1.2 Physical Demand.* Participants reported lower physical demand when using MAD (M = 2.00, SD = 1.27) and Source Code (M = 2.08, SD = 1.00) compared to Revela (M = 4.5, SD = 1.29) ($F_{(2, 69)} = 16.66$, $p < .001$). Post-hoc analyses showed significant differences between Revela and the other two conditions (both $p < .001$), with no significant difference between MAD and Source Code ($p = .98$).

*A.1.3 Temporal Demand.* Temporal demand scores were lower for MAD (M = 2.00, SD = 1.21) and Source Code (M = 2.41, SD = 1.72) than for Revela (M = 5.58, SD = 1.24), ($F_{(2, 69)} = 23.10$, $p < .001$). The post-hoc tests indicated significant differences between Revela and the other conditions (both $p < .001$), but not between MAD and Source Code ($p = .75$).

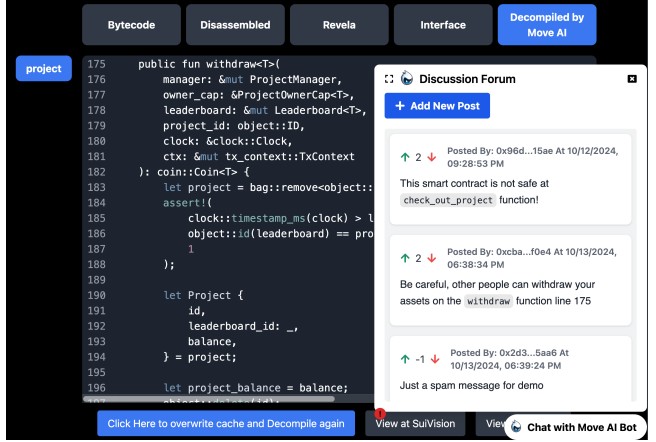

**Figure 7: Screenshot of the community forum, users can post their audit report here to point out where is the vulnerabilities to look into. Also users can vote to put spam messages down.**

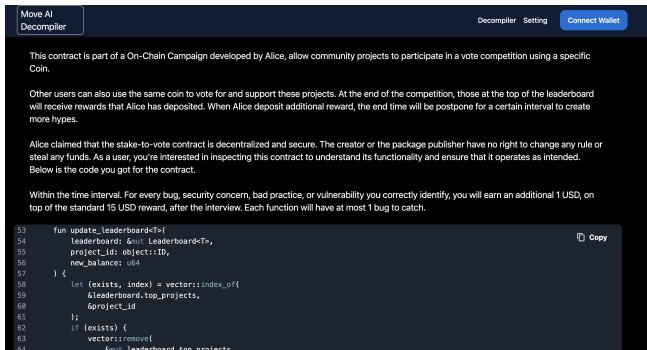

**Figure 8: The screenshot of the experiment interface, where users engaged with the task of coding understanding and vulnerability catching.**

*A.1.4 Effort.* Effort required was significantly less for MAD (M = 2.08, SD = 1.71) and Source Code (M = 2.41, SD = 1.73) compared to Revela (M = 5.25, SD = 1.71), ($F_{(2, 69)}$ = 14.97, $p$ < .001). Post-hoc comparisons showed significant reductions in effort for MAD and Source Code versus Revela (both $p$ < .001), with no significant difference between MAD and Source Code ($p$ = .86).

*A.1.5 Performance.* Performance scores were lower (indicating better perceived performance) for MAD (M = 2.08, SD = 1.24) and Source Code (M = 1.75, SD = 0.97) than for Revela (M = 4.33, SD = 2.02), $F_{(2, 33)}$ = 10.89, $p$ = .0002. Post-hoc comparisons showed significant differences between Revela and the Source Code ($p$ < .001) and MAD ($p$ = .002) condition, while no significant difference was found between MAD and Source Code ($p$ = .85).

*A.1.6 Frustration Level.* Participants experienced less frustration with MAD (M = 2.00, SD = 0.85) and Source Code (M = 1.67, SD = 0.78) compared to Revela (M = 5.42, SD = 1.73), ($F_{(2, 69)}$ = 35.85, $p$ < .001). Post-hoc analyses indicated significant differences between Revela and the other conditions (both $p$ < .001), but not between MAD and Source Code ($p$ = .78).

# B ADDITIONAL FIGURES

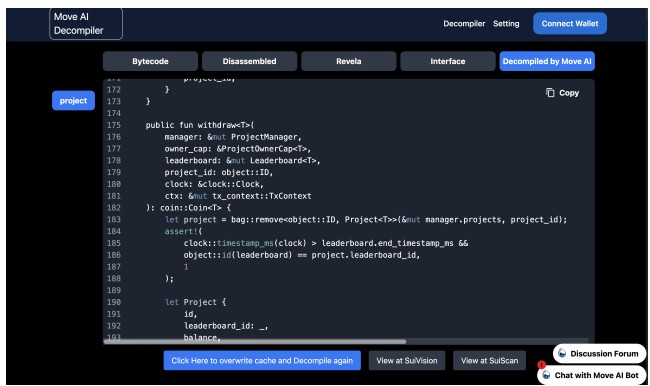

**Figure 5: The screenshot of the interface of Move AI Decompiler, where users can inspect different version of the code.**

```
public fun deposit_reward<T>(leaderboard: &mut Leaderboard<T>,
    // Implementation omitted for brevity
}

public fun withdraw_reward<T>(leaderboard: &mut Leaderboard<T>,
    // Implementation omitted for brevity
}

fun update_leaderboard<T>(leaderboard: &mut Leaderboard<T>, pro
    // Implementation omitted for brevity
}
```

**Figure 6: GPT's output if we send the whole smart contract code in the prompt and ask it to output the beautified version. We can see the function implementation is omitted and replaced with comments.**

