# OpenReview forum: "MAD: Move AI Decompiler to Improve Transparency and Auditability on Non-Open-Source Blockchain Smart Contract"
_ACM.org/TheWebConf/2025/Conference — WWW 2025 Oral_

### Official Review · Reviewer_n2hA · 2024-11-26

**Novelty:** 5
**Technical Quality:** 6

**Review:**

This paper introduces an AI-powered tool to decompile non-open-source Sui Move blockchain smart contracts into readable, recompilable source code. The tool can help address transparency issues in the Web3 ecosystem, allowing developers and auditors to verify the security of smart contracts. The approach (MAD) combines traditional decompilers with Large Language Models (LLMs). The authors show convincing evidence that the tool works, and a nice user study shows its potential.

Overall, I think this paper has its merits: 1) It is well-written and easy to read; 2) As someone who is not super familiar with decompilers, I found the work well situated in past work; 3) I found the system evaluation/user experiment convincing (although, the sample size is pretty small...).

The biggest weakness of this paper is that the key results are quantitative with N=5. This is just terribly underpowered! I don't know how hard it is to find MAD developers, but this would be a much, much stronger paper if this number was higher. Another potential criticism to the paper is that the whole thing is just a massive "ChatGPT" wrapper, a gigantic prompt that leverages the output of the Revela Decompiler and the move disassembler.  I don't think this is fair criticism, as unnecessary complexity is harmful: if this works, then great!

**Questions:**

- I didn't understand why, in 4.2, you mention 30 contracts, but in RQ1, you mention only 10. Is this because only 10 had unit tests?
- Is it super hard to recruit participants? Is that why n=5?

**Reviewer Confidence:**

2: The reviewer is willing to defend the evaluation, but it is likely that the reviewer did not understand parts of the paper

**Scope:**

4: The work is relevant to the Web and to the track, and is of broad interest to the community

---

### Official Review · Reviewer_iucK · 2024-11-30

**Novelty:** 4
**Technical Quality:** 4

**Review:**

Most frequent risk from Web3 is smart contract vulnerabilities, because smart contract usually published as bytecode, but the source code is not publicly accessible. This paper studies how to effectively decompile the bytecode into re-compileable source code.

This paper developed Move AI Decompiler (MAD).
1. Utilizes LLM (GPT-4o-2024-08-06)
2. Incorporates domain specific knowledge in the prompt, including: language features, syntax, variable mutation, object ownership.
3. Provides instructions regarding errors encountered in Revela's output.
4. Provides few-shot examples (17 samples)

Pros:
1. The paper is well written, easy to read and understand.
2. Recompile rate on both unit test and real-word case exceeds 60%
3. thorough user study was conducted to compare MAD with previous works, Revela. The study shows that codes produces by MAD are more human-readable  and require less mental effort to understand.

Cons:
1. The novelty is lacking. The problem formulation has been discussed in the previous works, and the solution is using LLM by utilizing hard-coded domain knowledge and few-shot prompting.
2. The paper could benefit from detailed comparative analysis with the previous works. In section 2.2, the authors mentioned Revela Decompiler and Move Dissambler, but comparative analysis hasn't been done to both works.
3. Since one of the contributions is the prompting technique, the paper could benefit from the ablation study.

**Questions:**

* On line 337, MAD splits chunks based on functions. If an LLM is fed a single function that calls another function, how can it consistently output the same name for the called function? (For instance, the function in the i-th chunk calls FuncA. FuncA is located in the (i+1)-th chunk. Since you fed these 2 chunks individually, how can LLM generate consistent name of FuncA in the i-th and (i+1)-th chunk?
* For the domain-specific knowledge, is the information static, or dynamic depending on the input? If dynamic, how do you generate such information?
* For the domain-specific knowledge, why did you only prompt based on the errors from Revela's output, and not also the Move Dissambler output?
* In section 5.1 and 5.2, only results from MAD is presented. What is the results for Revela and Move Dissambler?
* In section 6.1, what is the reason you were using different number of participants between Revela and MAD?
* In section 6.1, how many codes did each participant analyze, and were the codes coming from the same smart contract?
* In section 6.1, why you didn't have user studies with the Move Dissambler approaches?
* Can you do ablation to show the benefit for each component: Domain-specific knowledge, Should and should not instructions, and Few-shot examples?

**Reviewer Confidence:**

3: The reviewer is confident but not certain that the evaluation is correct

**Scope:**

3: The work is somewhat relevant to the Web and to the track, and is of narrow interest to a sub-community

---

### Official Review · Reviewer_3Lwx · 2024-12-03

**Novelty:** 6
**Technical Quality:** 4

**Review:**

The paper explores how LLMs may be applied to improve the readability of reverse-engineered smart contracts written in Move (as used by the Sui blockchain. A fairly small usability study was conducted finding that there is a significant improvement in readability, as compared to the state of the art reversed engineering tool (Revela). However, the accuracy of the tool is not particularly good, with the evaluation limited to unit tests and checking compilation. The expectation for a reverse engineering tool is that the output is functionally equivalent to the input, but the paper does not test for this. It also appears that there are frequent hallucinations which further decrease the expected accuracy.

Overall I think this is very interesting work, but seems to be at a relatively early stage of development both in implementation and evaluation. I would only expect this sort of system to be useful if it has a very high accuracy or at least will flag to the user if accurate decompilation fails.

**Questions:**

How can users of this tool obtain assurance that the output is accurate, or alternatively in which scenarios is it acceptable to have the level of accuracy achieved?

**Reviewer Confidence:**

3: The reviewer is confident but not certain that the evaluation is correct

**Scope:**

4: The work is relevant to the Web and to the track, and is of broad interest to the community